# Divergence of Intracellular Trafficking of Sphingosine Kinase 1 and Sphingosine-1-Phosphate Receptor 3 in MCF-7 Breast Cancer Cells and MCF-7-Derived Stem Cell-Enriched Mammospheres

**DOI:** 10.3390/ijms22094314

**Published:** 2021-04-21

**Authors:** Olga A. Sukocheva, Dong Gui Hu, Robyn Meech, Anupam Bishayee

**Affiliations:** 1Discipline of Health Sciences, College of Nursing and Health Sciences, Flinders University of South Australia, Bedford Park, South Australia 5042, Australia; 2Department of Clinical Pharmacology, College of Medicine and Public Health, Flinders University of South Australia, Bedford Park, South Australia 5042, Australia; donggui.hu@flinders.edu.au (D.G.H.); robyn.meech@flinders.edu.au (R.M.); 3Lake Erie College of Osteopathic Medicine, Bradenton, FL 34211, USA

**Keywords:** MCF-7 cells, breast cancer stem cells, sphingolipids, sphingosine kinase, sphingosine-1-phosphate receptor, estrogen, TNFα

## Abstract

Breast cancer MCF-7 cell-line-derived mammospheres were shown to be enriched in cells with a CD44+/CD24– surface profile, consistent with breast cancer stem cells (BCSC). These BCSC were previously reported to express key sphingolipid signaling effectors, including pro-oncogenic sphingosine kinase 1 (SphK1) and sphingosine-1-phosphate receptor 3 (S1P3). In this study, we explored intracellular trafficking and localization of SphK1 and S1P3 in parental MCF-7 cells, and MCF-7 derived BCSC-enriched mammospheres treated with growth- or apoptosis-stimulating agents. Intracellular trafficking and localization were assessed using confocal microscopy and cell fractionation, while CD44+/CD24- marker status was confirmed by flow cytometry. Mammospheres expressed significantly higher levels of S1P3 compared to parental MCF-7 cells (*p* < 0.01). Growth-promoting agents (S1P and estrogen) induced SphK1 and S1P3 translocation from cytoplasm to nuclei, which may facilitate the involvement of SphK1 and S1P3 in gene regulation. In contrast, pro-apoptotic cytokine tumor necrosis factor α (TNFα)-treated MCF-7 cells demonstrated increased apoptosis and no nuclear localization of SphK1 and S1P3, suggesting that TNFα can inhibit nuclear translocation of SphK1 and S1P3. TNFα inhibited mammosphere formation and induced S1P3 internalization and degradation. No nuclear translocation of S1P3 was detected in TNFα-stimulated mammospheres. Notably, SphK1 and S1P3 expression and localization were highly heterogenous in mammospheres, suggesting the potential for a large variety of responses. The findings provide further insights into the understanding of sphingolipid signaling and intracellular trafficking in BCs. Our data indicates that the inhibition of SphK1 and S1P3 nuclear translocation represents a novel method to prevent BCSCs proliferation.

## 1. Introduction

Sphingolipids, their receptors, and sphingolipid-metabolizing enzymes support and regulate the growth and survival of both normal and malignant cells. Sphingolipids are fundamental cell components that sustain the membrane barrier function, intracellular compartmentalization, and structural flexibility. Sphingolipids also operate as signaling molecules and control numerous biological processes including cell division, differentiation, migration, and survival; they have also been associated with malignant transformation [1,2,3]. Consequently, sphingolipid-modifying enzymes and receptors represent attractive therapeutic targets for a range of diseases, including cancer. Particularly, blocking and/or modifying agents that regulate sphingosine kinases (SphK1 and SphK2) and sphingosine-1-phosphate (S1P) receptor-signaling pathways have been intensively explored in breast cancer [3,4,5]. Notably, SphK2 was shown to transmit antiproliferative signals and its nuclear presence is well documented [6]. The signaling role of SphK1 is different and considered pro-oncogenic in breast cancers [2]. The nuclear presence of SphK1 remains questionable. Therefore, this study investigated growth- or apoptosis-triggered SphK1 trafficking in breast cancer cells.

Both SphK1 and SphK2 belong to the diacylglycerol kinase family and generate S1P from sphingosine [7]. S1P was shown to bind a family of G-protein coupled receptors, S1P1-5, linked to the control of cancer cell proliferation and metastasis [7,8]. Interestingly, SphK1 and SphK2 appear to have opposite effects in cancer: SphK1 is considered pro-oncogenic in breast cancers [2], while SphK2 was shown to facilitate antiproliferative signaling [6]. Although SphK1 was described as an enzyme mostly localized in cytosol [9], it was shown that SphK1 can be translocated to the plasma membrane [10] and nucleus [11]. Notably, immunohistochemical (IHC) staining of human cancer tissues indicated that the presence of SphK1 in nuclei is associated with shorter disease-specific survival and cancer recurrence [11]. Thus, the intracellular trafficking and localization of SphK1 may control its oncogenic properties.

The G-protein-coupled S1P receptor 3 (S1P3) is a membrane-localized protein, although the receptor is quickly internalized after ligand binding [12]. The internalized S1P3 can either be recycled or degraded in the lysosomes [13]. Considering the high concentration of S1P in blood plasma (approximately 1 μM), the dynamic transfer of S1P receptors between membrane and cytoplasm is a regular and continuous process. There is also evidence that this process can be stimulated by extracellular signals. For example, S1P3 activation and internalization was detected in response to treatment of MCF-7 cells with estrogen [1,12,14] and the selective estrogen receptor (ER) modulator tamoxifen [15]. However, the extent to which various growth-stimulating or -inhibiting agents control subcellular localization of SIP-signaling components, and the importance of this process in cancer, remains to be defined. Moreover, there are discrepant reports regarding the intracellular localization of both SphK1 and S1P3 from IHC staining of cancer tissues [11,16]. It is possible that the heterogeneous findings regarding SphK1 and S1P3 localization in cancer are reflective of underlying cellular heterogeneity. In particular, tumors generally contain a subpopulation of relatively undifferentiated cells with progenitor-like characteristics/markers, an intrinsic self-renewal capacity, and a remarkable resistance to apoptosis: these are termed cancer stem cells (CSC).

Breast CSC (BCSC) were shown to employ the SphK1-S1P signaling axis to maintain their stem-like phenotype and initiate metastasis [17,18]. S1P3 knockdown reduced the BCSC population [18], while the activation of S1P pathway boosted the capacity of BCSCs to form mammospheres and triggered Notch signaling, which is an important pathway for self-renewal [17]. Recent studies demonstrated a link between the SphK1/S1P axis and downregulation of STAT1 activity in stem-cell-enriched mammospheres [19], supporting the existence of a sphingolipid-dependent mechanism for the control of transcription factor activity in CSC. Given the capacity of SphK1 to localize to the nucleus, it is possible that SphK1 directly acts as a transcriptional co-regulator.

Subcellular localization can regulate protein function by controlling the repertoire of interaction partners; as an example, proteins that function as enzymes in one subcellular compartment/organelle may have non-enzymatic functions in another organelle. This is likely to be an important mechanism for diversifying protein functions. During development and histogenesis, protein localization may be regulated by extrinsic signals from the surrounding environment, and intrinsic factors including the extent of cellular differentiation. This can be important for normal tissue physiological function. Thus, knowledge of protein localization and transfer during signaling may contain valuable information related to enzyme/receptor function. Moreover, aberrant localization of proteins has been linked to anomalous signaling in a variety of pathogenic contexts, including metabolic, cardiovascular, and neurodegenerative abnormalities, as well as cancer [20]. With respect to the SphK-S1P signalling axis, it is possible that nuclear translocation of signalling components such as SphK1 and S1P3 in cancer cells facilitates pathogenic progression. As such, blockade of SphK1 and S1P3 nuclear translocation could potentially be a novel approach to control cancer cell proliferation through blockade of downstream gene expression programs.

The aim of the current study was to characterize and compare the effects of specific SphK1-activating agents on the intracellular localization of SphK1 and S1P3 in ER-positive MCF-7 breast cancer cells and MCF-7-derived, BCSC-enriched mammospheres. We questioned whether apoptosis-inducing agents can influence the intracellular trafficking of SphK1 and S1P3 in cancer cells. Considering that compartmentalization of the protein defines the range of its potential signaling partners and mechanism of action, the purpose of this study was to visualize the localization of SphK1 and S1P3 in cells treated by SphK1- and S1P3-activating substances with growth- or apoptosis-promoting effects. Protein localization was confirmed using both immunofluorescent (IF) staining [12,21] and cell fractionation [22,23] techniques. The findings indicate that S1P and estrogen stimulate translocation of SphK1 and S1P3 into the MCF-7 cell nuclei. Moreover, IF analysis of MCF-7-derived BCSC-enriched mammospheres revealed the high heterogeneity of both SphK1 and S1P3 expression and localization.

## 2. Results

### 2.1. Responses of Parental MCF-7 Cells and MCF-7-Derived Mammospheres to TNFα

MCF-7 cells were used to generate CD44+/CD24− breast cancer stem cell (BCSC)-enriched mammospheres (Figure 1A,B) following previously reported methods [17,18,19,23]. The proportion of cells with the CD44+/CD24− BCSC marker profile in mammospheres was 18.4 ± 0.17%, which was significantly higher than that in the parental cell line (0.31% ± 0.01) (Figure 1A). The BCSC-enriched mammosphere culture was sensitive to pro-apoptotic cytokines (INFγ and TNFα) treatment, which decreased mammosphere growth (Figure 2A,B). The level of S1P3 was significantly higher in mammospheres compared to parental cells; however, this was reduced after INFγ and TNFα treatment (Figure 2C). INFγ and TNFα also induced apoptosis in the parental MCF-7 cell line (Figure 2D).

### 2.2. Regulated Internalization and Nuclear Translocation of SphK1 and S1P3 in MCF-7 Cells

To identify potential differences in sphingolipid signaling between BCSC-enriched and parental MCF-7 cells, we first visualized the localization of SphK1 and S1P3 in parental MCF-7 cells using IF with commercially available antibodies and confocal microscopy as described previously [12,15]. SphK1 was localized in the cytoplasm of parental vehicle-treated (control) cells (Figure 3A,B). Some heterogeneity of the enzyme expression level was apparent and might be associated with different stages of cell growth and division. The pattern of SphK1 localization was altered after treatment of cells with the SphK1-activating estrogen and with the receptor agonist S1P. Specifically, perinuclear and nuclear localization of SphK1 (observed as the overlap of Hoechst stain with Alexa-488-labelled endogenous SphK1 immunofluorescence) was detected in a subpopulation of cells after treatment with estrogen or S1P for 3 h (Figure 3). Although S1P promoted SphK1 nuclear localization, TNFα, which is known to stimulate the production of S1P, did not induce SphK1 nuclear translocation [24]. Instead, TNFα treatment induced MCF-7 cell rounding, anoikis, and membrane blebbing consistent with the activation of apoptosis by this pro-inflammatory cytokine. While the nuclear localization of SphK1 in TNFα-treated cells was not observed, the presence of the enzyme in proximity to the plasma membrane and in cytosol was detected. Analysis of co-localization of both Alexa-488 (green)-labelled SphK1 and S1P3 and Hoechst (blue nuclei) is shown in Figure 3B.

S1P3 localization was also examined in parental MCF-7 cells. A fraction of S1P-treated cells demonstrated nuclear staining for S1P3, although there was a decrease in overall endogenous S1P3 immunofluorescence during S1P treatment (Figure 4). Consistent with our previous findings [12], S1P3 was internalized after 3 h exposure to estrogen. Moreover, in a large subpopulation of cells, most of the internalized receptor was located in the nuclear space (Figure 3B and Figure 4). TNFα treatment did not stimulate the translocation of S1P3 into the nuclear space, although the decrease in the total immunofluorescence suggested S1P3 degradation.

The confocal imaging analysis was confirmed using cell fractionation experiments (Figure 5). Subcellular fractions were characterized using specific subcellular compartment markers. Nucleostemin was used as a marker for the nuclear fraction, Na+/K+-ATPase for the membrane fraction, and calpain-1 for the cytoplasm fraction. Results showed (Figure 5A,B) that SphK1 and S1P3 endogenous proteins are localized to both the membrane and cytosol fractions in control cells. Estrogen and S1P treatments increased S1P3 internalization and the nuclear pool of SphK1 and S1P3 (Figure 5C). In contrast to the effects of estrogen and S1P, TNFα stimulated degradation of S1P3 but did not induce nuclear localization of SphK1 and S1P3. These findings are consistent with the results from imaging analysis.

### 2.3. Localization of SphK1 and S1P3 in BCSC-Enriched Mammospheres

Hirata and colleagues [17] detected enhanced S1P3 expression and progenitor cell-related functioning in BCSCs derived from MCF-7 cells. However, S1P3 subcellular localization was not examined in this context. We assessed SphK1 and S1P3 localization in mammospheres using IF and confocal microscopy (Figure 6). BCSC-enriched mammospheres demonstrated highly heterogenous SphK1 and S1P3 expression and localization in the absence of any treatment (control cultures). Hence, it was not possible to conclusively determine the effects of S1P or estrogen on these factors in mammospheres (data not shown). Live cell fluorescent monitoring of single-cell-based changes may help to spot the difference in future experiments. There were no differences in S1P3 trafficking in mammosphere cells, although the level of S1P3 expression was higher compared to parental MCF-7 wild-type cells. The increased expression was supported by RT-PCR analysis of S1P3 mRNA levels, suggesting that it occurs, at least in part, via a transcriptional mechanism (Figure 2C). S1P3 degradation with no nuclear translocation was commonly observed in the majority of TNFα-treated mammospheres (Figure 6). Interestingly, TNFα stimulated SphK1 nuclear translocation in a sub-population of mammosphere cells (Figure 6), which contrasted with its effects on parental MCF-7 cells.

## 3. Discussion

In this study, we demonstrated that SphK1, located mostly in the cytoplasm of untreated parental MCF-7 cells, translocates to perinuclear and nuclear spaces in a subpopulation of cells after treatment with pro-proliferative agents that directly or indirectly activate the S1P signaling pathway (i.e., S1P or estrogen). In contrast, treatment with pro-apoptotic agent (TNFα) did not lead to SphK1 nuclear accumulation in parental MCF-7 cells. However, in BCSC-enriched mammospheres, TNFα stimulated the cytoplasm-to-nucleus translocation of SphK1 in a subset of cells. Mammosphere cells demonstrated enhanced expression of S1P3 as compared to MCF-7 parental cells. TNFα induced apoptosis and S1P3 degradation with no nuclear translocation of the receptor in both parental MCF-7 and mammosphere cells. Similar to their effects on SphK1, S1P and estrogen stimulated nuclear translocation of S1P3. A summary of our findings is depicted in Figure 7.

The level of SphK1 nuclear localization was similar between S1P- and estrogen-treated cells, where the enzyme was mainly observed in the nuclei of smaller cells which may have recently undergone division. Activation of the SphK1/S1P signaling axis was previously associated with intracellular trafficking of SphK1 and S1P3 proteins, followed by diverse biological responses. Previous immunohistochemical analysis of paraffin-embedded breast cancer tissues and cells indicated heterogeneous SphK1 and S1P3 nuclear localization [11,16]. S1P/estrogen-induced cancer cell proliferation was reported previously [3,5,14,25]. Estrogen, a critical growth-stimulating and pro-survival agent in ER-positive breast cancer cells, is known to activate proliferation-related intracellular effectors including Erk1/2 [26], PI3K/Akt [27], and SphK1 [1,28]. S1P was also shown to stimulate proliferation in different cancer cell contexts [2,3]. Both S1P and estrogen can stimulate cell division via transactivation of the EGF receptor and internalization of S1P3 in MCF-7 cells [1,2,12], although other mechanisms were also reported [25]. Erk1/2 activation increases the overall level of protein phosphorylation, and the phosphorylation of SphK1 can promote its trafficking from cytosol to the plasma membrane [10]. However, our study did not detect trafficking of SphK1 to the plasma membrane in estrogen- or S1P-treated MCF-7 cells. One limitation of the current study is that SphK1 phosphorylation was not examined. Co-staining of plasma-membrane-localized and phosphorylated SphK1 and live cell imaging might help to clarify the effects of S1P signaling activators on plasma membrane trafficking of SphK1 in future studies. However, previous work indicates that membrane-associated SphK1 has a short half-life (and, hence, low steady-state abundance), leading to technical difficulties with quantitative analysis [29,30,31,32].

In this study, we examined the localization of SphK1 but not SphK2 in MCF-7 cells. This was, in part, because the functional roles of nuclear SphK2, including generation of nuclear S1P [33,34], and the mechanisms controlling its nuclear import and export have already been extensively studied [34,35,36]. In contrast, the function of nuclear SphK1, and the mechanisms controlling its nuclear translocation, remain less well-defined. Nuclear localization of SphK1 has been observed in breast cancer cells, and greater nuclear SphK1 levels have been linked to advanced cancer prognosis, significantly shorter disease-specific survival, and faster cancer recurrence [11]. Immunohistochemical analysis of mammary tumors indicated that nuclear SphK1 can co-localize with Erk1/2, Lck/Yes novel tyrosine kinase (LYN), V-akt murine thymoma viral oncogene homolog (AKT), or Nuclear Factor κB (NF-κB). Moreover, the SphK1/SphK2 product, S1P was shown to target histone deacetylases (HDAC1/2), tumor necrosis factor receptor associated factor 2 (TRAF2), human telomerase reverse transcriptase (hTERT), and NF-κB nuclear activities linked to cancer initiation and progression [24,33,34,37,38,39]. SphK1 activation was also recently observed to suppress the expression and phosphorylation of signal transducer and activator of transcription 1 (STAT1), which is an INFγ signaling-linked transcription factor [19,40]. Our demonstration that pro-proliferative signaling promotes SphK1 nuclear-localization in MCF-7 cells prompts further analysis of its cell-type-specific functions in breast cancer.

SphK1 contains two functional nuclear export signal (NES) sequences, proposed to control shuttling between the cytoplasm and nucleus [41]. The process is leptomycin B-sensitive, indicating that it is dependent on chromosome region maintenance 1 (CRM1) [41,42]. However, the detailed mechanisms regulating nuclear translocation of SphK1 remain to be defined [30]. Previous studies indicated a striking co-localization of endogenous SphK1 with centrosomes in HEK293 cells [31]. However, in general, where SphK1 locates within the nucleus and the identity of its nuclear interactors are also poorly defined. While our studies support extrinsic signal-regulated translocation of SphK1 to the nucleus in MCF-7 cells, differential mechanisms controlling this process in non-stem breast cancer cells and BCSCs require further investigation. Similarly, the mechanisms by which SphK1 may control gene expression in these different contexts requires more study. It is possible that the generation of S1P and S1P-derived long-chain fatty aldehydes is involved in epigenetic gene regulation during the adaptation of cancer cells to a cytokine-mediated hostile microenvironment, as was shown during lung inflammation [43]. However, this idea requires testing. The apparent enhancement of nuclear SphK1 in recently divided cells requires further confirmation using specific cell-cycle markers.

The intracellular localization patterns of the S1P receptors were recently assessed in human tissues. Employing tissue IHC microarrays, Wang and colleagues [16] analysed 384 formalin-fixed paraffin-embedded samples containing benign and malignant tissues. It was found that all five S1P (1–5) receptors are widely distributed in multiple human tissues/cell types (albeit with varying abundance) and are located in both the cytoplasm and nucleus. In the present study, only S1P3 trafficking was studied in MCF-7 cells, largely because other SIP receptors show very low expression in this model (often below detection), and also because they have not been directly associated with breast cancer progression. We observed S1P3 internalization and nuclear localization in subpopulations of cells treated with S1P or estrogen. Supporting our findings, another group observed S1P3 localization only in the nucleus [11]. A decrease in total fluorescence indicated degradation of the S1P3 receptor after treatment with TNFα, which did not induce translocation of S1P3 to the nuclear space. How S1P3 is translocated to the nucleus and what its nuclear function is remain to be understood. A possibility of intracellular trafficking of other S1P receptors cannot be excluded.

It is somewhat surprising that S1P receptors are hard to observe at the cellular membranes, given that S1P receptors belong to the transmembrane G-protein-coupled receptor (GPCR) class [4,8]. However, along with S1P3 [11], several other GPCRs were recently found in the nuclear space, suggesting novel roles for this type of receptor beyond those traditionally established [44]. Distinct F2rl1 (formerly known as PAR2) roles were shown to be dependent on the receptor subcellular localization at the plasma membrane or at the nucleus [44]. Several mechanisms of nuclear GPCR translocation have been reviewed recently [45]. However, specific physiological functions of nuclear S1P receptors remain to be discovered. Contrary to the effect of S1P/estrogen, TNFα did not induce nuclear localization of SphK1, although pro-inflammatory cytokine TNFα was shown to stimulate the production of S1P by SphK1 in other immortalized cells [24,46,47]. The effect of TNFα on the enzymatic activity of SphK1/2 in MCF-7 breast cancer cells was never tested comprehensively. However, it was shown that TNFα can downregulate the expression of SphK1 via the lysosomal release of cathepsin B and degradation of the enzyme in MCF-7 cells [48]. Furthermore, S1P/estrogen and TNFα induced the opposite biological effects in MCF-7 breast cancer cells. While S1P stimulates proliferation [1,3,10], the cytokine treatment induces apoptosis in breast cancer cells [49]. Accordingly, MCF-7 cell rounding, apoptosis, anoikis, and membrane blebbing were observed in our experiments with TNFα. Secreted by tumor infiltrating macrophages (tumor microenvironment), TNFα was suggested to activate SphK1, stimulate the expression of S1P3, and promote the development of cancer cell resistance and metastasis in a subset of TNFα-resistant cells [17,50]. However, our study is the first to report intracellular trafficking of SphK1 in MCF-7 wild-type cells and BCSCs exposed to TNFα. Given that BCSCs are generally apoptosis-resistant, this finding warrants further investigation, as it may suggest that BCSC have unique mechanisms that allow nuclear translocation of SphK1 in the face of pro-apoptotic signaling.

Enhanced S1P3 expression was reported in BCSC-enriched mammospheres previously [17,18] and confirmed in our experiments. Furthermore, MCF-7-derived mammosphere cells demonstrated high levels of heterogeneity of SphK1 expression and localization in our experiments. Mammospheres form from single cells that have the capacity to self-renew in anchorage-independent, serum-derived conditions (i.e., show a tumor initiating capacity). However, as spheres expand the progeny of these tumor-initiating cells, they typically manifest varying degrees of stemness/differentiation, which can potentially generate a large variety of responses. The cellular heterogeneity of mammospheres is an inherent limitation and was underscored by our detection of the CD44+/CD24- marker profile in less than 20% of the mammosphere cells. High levels of phenotypic and genetic heterogeneity in breast tumors, including varying abundance of stem-like cells, were reported in numerous studies [51,52,53,54]. Accordingly, it is not unexpected that heterogeneity will be observed in SphK1 localization in mammospheres. Future work could use live cell imaging and/or cell sorting of mammosphere subpopulations, to characterize SphK1 and S1P receptor expression and localization within different cell sub-populations. This approach could also permit the analysis of time- and dose-dependent responses of SphK1 localization in progenitor/BCSCs cells exposed to various SphK1 activating or inhibiting agents.

The specific localization and compartmentalization of proteins facilitates interactions with effectors and downstream targets, thus directing signaling outcomes. It has been suggested that disease-related subcellular mis-localization of proteins is an attractive target for therapeutic interventions and a strategy to inactivate disease-causing proteins [20]. Recent evidence indicates that the functionality of key growth-regulating proteins can depend on subcellular location. For instance, the EGF receptor undergoes translocation into different organelles, eliciting distinct functions in response to different stimuli, including endogenous ligands, radiation, and targeted anticancer therapy [53]. A classic example of the therapeutic relevance of protein localization is estrogen receptor nuclear localization, which is a biomarker of response to anti-hormonal therapy in breast cancer patients worldwide [54]. Whether nuclear localization/mis-localization of S1P signaling components in cancer can serve as a prognostic or predictive biomarker, or a novel target for drug development, is an important question for future research.

In conclusion, the current study detected differences in SphK1 and S1P3 localization in parental and MCF-7-derived, BCSC-enriched mammosphere cells treated with growth- and apoptosis-inducing agents. Both parental and BCSC-enriched MCF-7 cultures were sensitive to the pro-apoptotic effects of TNFα, which prevented the nuclear translocation of S1P3. This observation might serve as a diagnostic marker of sensitivity to apoptosis in breast cancer cells. It also suggests a testable hypothesis that apoptosis-resistant cells may develop mechanisms to facilitate nuclear translocation of SphK1, despite the activation of pro-apoptotic signaling. Moreover, our findings provide the groundwork for investigation into SphK1 and/or S1P3 nuclear translocation as a novel therapeutic target in BCSC.

## 4. Materials and Methods

### 4.1. Chemicals, Cell Culture, and Antibodies

Sphingosine-1-phosphate (S1P) was purchased from Biomol Research Laboratories Inc. (Plymouth Meeting, PA, USA). S1P was dissolved in 10 mM NaOH at 3 mM as described previously [1,12,14]. Estrogen (17β-estradiol) and TNFα were bought from Merck (Bayswater, VIC, Australia). The human breast cancer cells MCF-7 cells were purchased from ATCC^®^ (Manassas, VA, USA) and cultured in phenol red-free Dulbecco’s modified Eagle’s medium (CSL Biosciences, Parkville, Australia) supplemented with 10% fetal bovine serum. Culture medium contained L-glutamine (2 mM), nonessential amino acids, penicillin (100 U/mL), and streptomycin (100 mg/mL) as described [12,14,15].

Fluorescein isothiocyanate (FITC)-conjugated monoclonal anti-CD24 (cat. #555427) or phycoerythrin (PE)-conjugated anti-CD44 (cat. #550989) reagents were purchased from BD Biosciences (Franklin Lakes, NJ, USA).

### 4.2. Generation and Maintenance of MCF-7 Cancer Stem Cell-Enriched Mammospheres

It was reported that CD44+/CD24– breast cancer cell fraction is enriched with cells with stem-cell-like properties (BCSCs) [55]. The mammosphere growth assay is a method to enrich and propagate CD44+/CD24- BCSCs in three-dimensional (3D) culture, as described previously [17,56,57]. Following this technique, MCF-7 cells were grown as mammospheres in serum-free medium containing 10 ng/mL basic fibroblast proliferation factor, 20 ng/mL epidermal growth factor (EGF), and 2% B-27 to induce the mammosphere formation for 14 days [17,57]. Mammospheres were enriched in cells with the CD44+/CD24+ marker profile, reaching nearly 20% (Figure 1), as verified by flow cytometry as described previously [58,59]. Mammospheres images were captured and counted using an IX71 microscope at ×400 (Olympus, Notting Hill, VIC, Australia).

### 4.3. Flow Cytometry Assays: Apoptosis and CD44+/CD24- Marker Analysis

Annexin V/propidium iodide (PI) (Cayman Chemicals, Ann Arbour, MI, USA) flow cytometry analysis was used to assess the level of apoptosis in parental MCF-7 cells as described previously [1]. Flow cytometry was also used to estimate percentages of CD44+/CD24- cells in parental and disaggregated MCF-7-derived mammospheres, [17,19]. Briefly, single cell suspensions (1 × 10^6^ cells/mL) were incubated in the dark at 4 °C for 30 min with FITC-conjugated anti-CD24 (1:400) followed by PE-conjugated anti-CD44 (1:400) in staining buffer (3% FBS + 0.01% sodium azide). Following staining, the cells were washed twice with PBS (Sigma-Aldrich, Castle Hill, NSW, Australia) and analysed on a BD FACS Calibur flow cytometer (BD Biosciences, North Ryde, NSW, Australia).

### 4.4. Subcellular Fractionation and Immunoblotting

After incubation with specified agents (10 nM E2, or 500 nM S1P, or 100 ng/mL TNFα for 3 h), cells were harvested, washed with PBS, and lysed in buffer containing protease inhibitors (Sigma-Aldrich, Castle Hill, NSW, Australia). Membrane, cytoplasm, and nuclear fractions were extracted using the Qproteome cell compartment kit, corresponding to the manufacturer’s instructions (Qiagen, Chadstone Centre VIC, Australia) that were developed according to the previously described technique [22]. The membrane fraction contains endosomes and membrane-compartmented organelles and cell surface plasma membrane [23]. The protein concentrations of each fraction were determined using the BCA protein assay kit (Pierce Biotechnology, Rockford, IL, USA). Aliquots of total cell fractions with equal amounts of total protein (50 μg) were resolved on 10% SDS-polyacrylamide gel and transferred to Hybond-P membranes (Sigma-Aldrich, Castle Hill, NSW, Australia). The membranes were probed with the following antibodies: anti-SphK1 (1:1000; polyclonal Abs cat# 12071, Cell Signaling Technologies, Arundel, QLD, Australia) [60], anti-S1P3 (1:500, Sapphire Biosciences Pty Ltd., Redfern NSW, Australia; Cayman Chemical (Ann Arbour MI, USA) polyclonal Abs cat #10006373) [15], anti-Na^+^/K^+^-ATPase (sc-48345, Santa Cruz Biotechnology, Inc., Dallas, TX, USA) (plasma membrane marker), anti-calpain-1 (ab28258, Abcam, Cambridge, MA, USA) (cytoplasmic marker), and anti-nucleostemin (ab70346, Abcam, Cambridge, MA, USA) (nuclear marker). Signals were revealed by enhanced chemiluminescence (ECL Plus kit (GE Healthcare, Parramatta, Australia)). ImageQuant 350 software (Molecular Dynamics, Sunnyvale, CA, USA) was used to analyse the relative amount of protein.

### 4.5. Immunofluorescent (IF) Analysis Using Confocal Microscopy

SphK1 and S1P3 expression and cellular localization were visualized using confocal microscopy. MCF-7 parental cells were seeded onto Fibronectin (Invitrogen, Carlsbad, CA, USA) coated glass slides (Lab-Tech, Naperville, IL, USA) [12] and cultured with indicated agents (10 nM E2, or 500 nM S1P, or 100 ng/mL TNFα for 3 h). CSC-enriched MCF-7 mammospheres were collected by centrifugation in a Cytospin (Thermo, Waltham, MA, USA), and seeded on the coated glass slides prior to fixation. After stimulation with agents or vehicle controls, cells were washed twice and fixed with 4% paraformaldehyde and permeabilized in 0.1% Triton X-100. After washing with PBS, blocking buffer (10% normal serum/PBS) was applied to prevent nonspecific staining. The samples were incubated with specific antibodies, as described previously [12,15], or according to manufacturer instructions. Slides were then washed three times with PBS and incubated for 1 h at room temperature with secondary antibodies conjugated to fluorescent fluorophores (Alexa Fluor 488, Molecular Probes, Eugene OR, USA) as described previously [12,21]. Nuclei were stained with Hoechst dye according to the protocol recommended by the manufacturer. Fluorescence images were analysed using a Leica TCS SP5 Scanning Confocal microscope and Leica LAS AF software.

### 4.6. RT-PCR Analysis

Total RNA was extracted from cell cultures using TRIzol according to the manufacturer instructions (Invitrogen, Carlsbad, CA, USA), and then reversely transcribed to prepare cDNA using the PrimeScript RT Reagent kit (TakaRa, Otsu, Shiga, Japan). The mRNA levels of SphK1 and S1P3 were subsequently detected by PCR with an internal β-actin control. The primers used to amplify were as follows: SphK1 sense 5′-TTGAACCATTATGCTGGCTATGA and antisense 5′-GCAGGTGTCTTGGAACCC; S1P3 sense 5′-GCCCTCTCGTGGATTTTGG and antisense 5′-CGCATGGAGACGATCAGTTG.

### 4.7. Statistical Analysis

Results from three to four independent experiments were pooled and presented as mean values ± standard deviation (SD). The significance of differences was determined using Student’s *t*-test or one-way analysis of variance (ANOVA) tests using SigmaPlot 12.0 software package (Systat Software Inc, San Jose, CA, USA). Quantification of the mean fluorescence intensity and co-localization in selected image regions was performed by using the ImageJ software (http://rsb.info.nih.gov/ij/, accessed on 1 November 2015). Localization of SphK1 and S1P3 in the nuclei is presented as a percentage of the overlapping green and blue fluorescence compared to the total green fluorescence per selected area of an image (Figure 3B). Five random fields per one image, from at least three independent experiments, were examined. Differences were considered statistically significant at a level of *p* < 0.05 or <0.01.

## Figures and Tables

**Figure 1 ijms-22-04314-f001:**
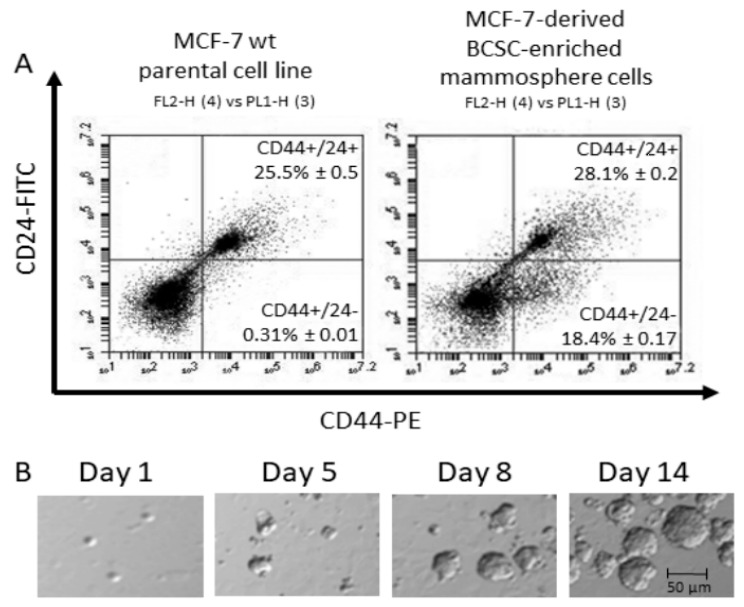
Phenotypic characterization of MCF-7-cell-derived mammospheres. (**A**) Flow cytometry was used to estimate the percentage of CD44+/CD24– cells (mean ± SD from three independent experiments) in parental MCF-7 cultures (monolayer) and MCF-7-derived mammosphere cultures. (**B**) Phase-contrast microscopy was used to monitor mammosphere growth over 14 days. Images are representative of at least 3 independent experiments.

**Figure 2 ijms-22-04314-f002:**
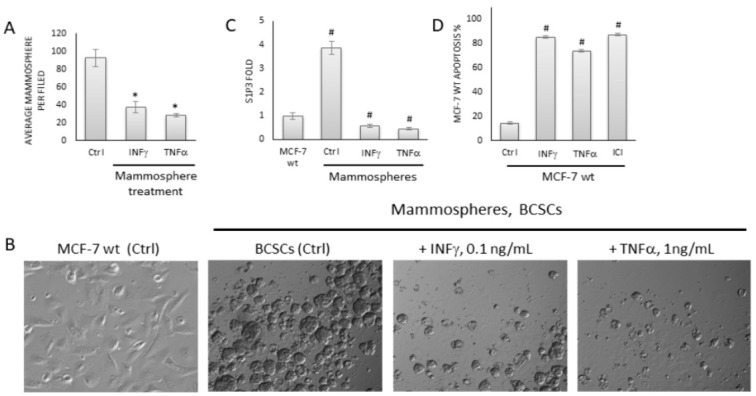
Biological effects of TNFα on parental MCF-7 cells and MCF-7-derived, BCSC-enriched mammospheres. (**A**). TNFα and INFγ treatments led to significant reduction in mammosphere growth. Representative images are shown (×20); experiments were repeated at least 4 times with similar results. (**B**). MCF-7 cells (Ctrl; controls) are shown as an example of parental cells line phenotype. Mammospheres were treated with TNFα and INFγ and photographed. Representative images are shown. (**C**) Rt-PCR analysis of S1P3 expression in parental and MCF-7-derived mammospheres. The level of housekeeping gene β-actin was used to estimate fold change in S1P3 expression. (**D**) INFγ (0.1 ng/mL), TNFα (1 ng/mL), and ICI182780 (ICI; 10 nM) stimulated apoptosis in MCF-7 wild-type (parental) cells after 72 h treatment. Flow cytometry was used to estimate apoptosis % of PI and Annexin V positive cells. * *p*-value < 0.05; # *p*-value < 0.01.

**Figure 3 ijms-22-04314-f003:**
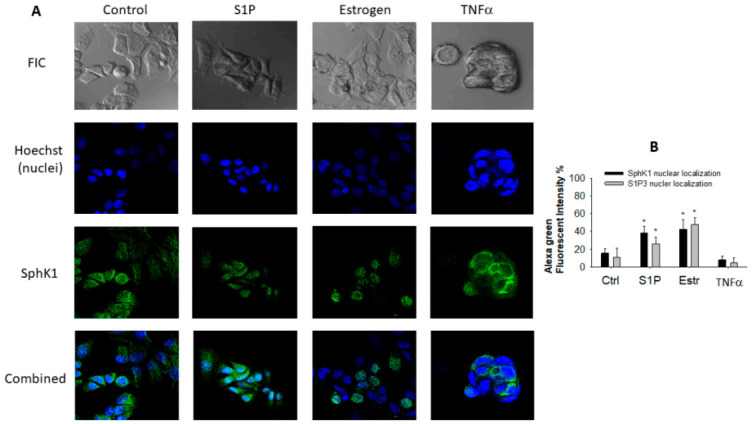
SphK1 intracellular localization in parental MCF-7 cells. (**A**) SphK1 intracellular localization was assessed using IF and confocal microscopy in cells treated with vehicle (Ctrl), or treated with 10 nM Estrogen, 500 nM S1P, or 100 ng/mL TNFα for 3 h. The 3 h timepoint was optimized in pilot studies to identify significant effects on trafficking. Endogenous SphK1 protein (green) was visualized using Alexa 488-conjugated secondary antibodies. Hoechst (blue) was used to mark cell nuclei. (**B**) Alexa-488 (green) and Hoechst (blue) co-localization percent was estimated in MCF-7 cells treated as described in the Methods (* *p*-value < 0.05). S1P3 nuclear localization was estimated using IF imaging as shown in Figure 4. Total Alexa-488 fluorescence intensity per analysed image area was taken as 100%. Images are representative of at least 3 independent experiments.

**Figure 4 ijms-22-04314-f004:**
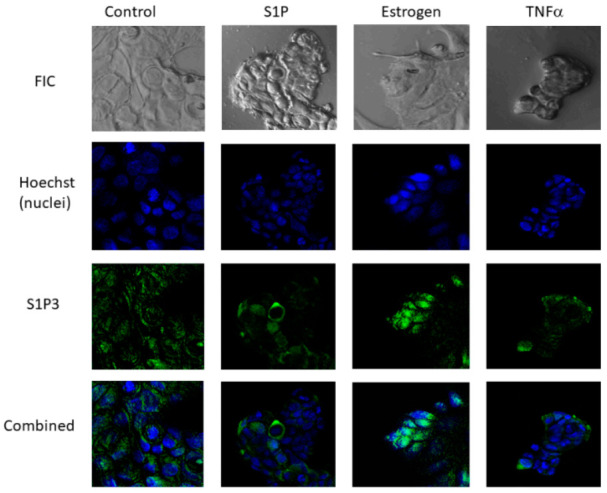
S1P3 intracellular localization was visualized using IF and confocal microscopy (×400) in parental MCF-7 cells treated with vehicle (control), 10 nM estrogen, 500 nM S1P, or 100 ng/mL TNFα for 3 h. Endogenous S1P3 protein (green) was visualized using Alexa 488-conjugated secondary antibodies. Representative images are shown; experiments were repeated three times.

**Figure 5 ijms-22-04314-f005:**
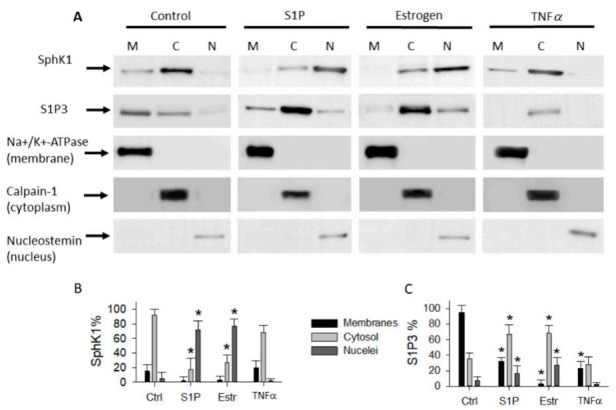
Immunoblot analysis of SphK1 and S1P3 localization in parental MCF-7 subcellular fractions. (**A**) Specific marker proteins were used to evaluate subcellular fractionation and gel loading: nucleostemin was used as marker for nuclear fraction (N), Na+/K+-ATPase for membrane fraction (M), and calpain-1 for cytoplasm fraction (**C**). Parental MCF-7 cells were treated as described in Figure 1. (**B**) Semi-quantitative analysis of SphK1 and S1P3 levels using gel densitometry. The SphK1 level in cytosol and S1P3 level in membranes of control cells were set as 100%. Results are the means of three independent experiments ±SD. Significance of the differences (* *p* < 0.05) was assessed between control and agent-induced effects. Images are representative of at least 3 independent experiments.

**Figure 6 ijms-22-04314-f006:**
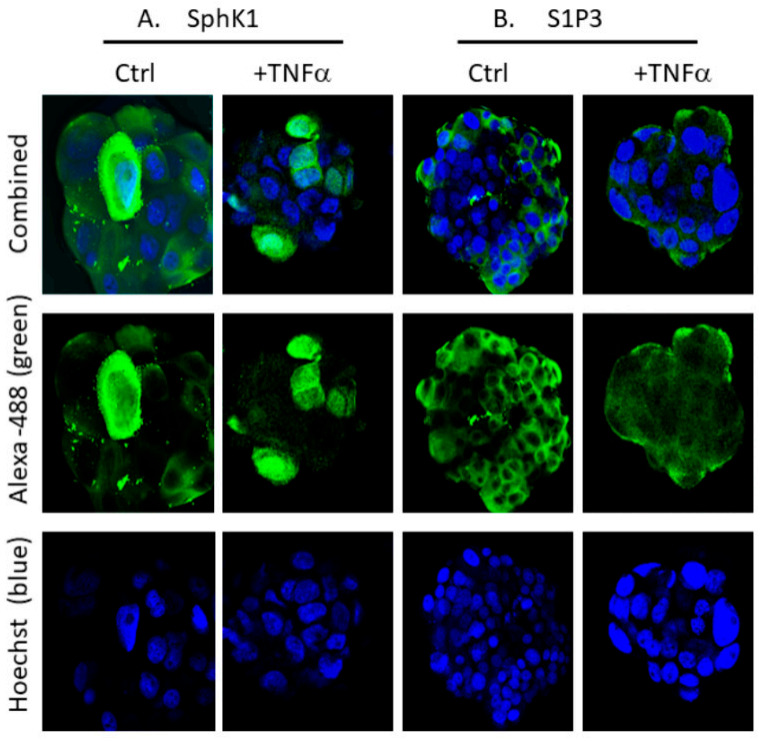
Localization of SphK1 (**A**) and S1P3 (**B**) was visualized in BCSC-enriched mammospheres using confocal microscopy (×400). (**A**). Heterogeneous SphK1 expression and localization (in cytoplasm) was observed in vehicle-treated mammosphere cells (Ctrl). Nuclear SphK1 localization was observed in TNFα-treated cells, although the response was heterogenous. (**B**). No nuclear localization of S1P3 was observed in TNFα-treated cells. TNFα reduced S1P3 membrane localization and overall fluorescence compared to Ctrl. Images are representative of at least 3 independent experiments.

**Figure 7 ijms-22-04314-f007:**
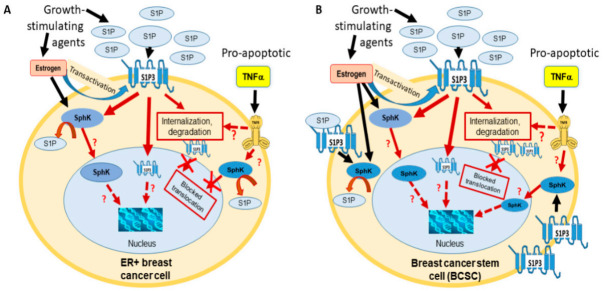
Schematic presentation of SphK1 and S1P3 trafficking in parental MCF-7 cells (**A**) and MCF-7 derived BCSC-enriched mammospheres (**B**). Increased S1P3 expression and transformed TNFα signaling was detected in BCSC-enriched mammospheres (**B**).

## Data Availability

The data presented in this study are available in this article.

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
