# Peer review of "Divergence of Intracellular Trafficking of Sphingosine Kinase 1 and Sphingosine-1-Phosphate Receptor 3 in MCF-7 Breast Cancer Cells and MCF-7-Derived Stem Cell-Enriched Mammospheres"

_ijms, 2021, doi:10.3390/ijms22094314_

Round 1

Reviewer 1 Report

Manuscript by Olga A. Sukocheva et al. reports on comparative analysis of S1P-producing sphingosine kinase 1 (SphK1) and S1P receptor 3 (S1P3) in breast cancer MCF-7 cell line and MCF-7-derived mammospheres upon treatment with S1P and estrogen as growth-stimulating agents and TNFalpha as pro-apoptotic agent. The focus was given to phenomenon of intracellular trafficking of sphingolipid-related molecules such as triggered translocation of SphK1 and S1P3 from the cytoplasm to the nuclei. Summarizing the study outcome, the authors conclude that the inhibition of nuclear translocation may represent novel way to prevent proliferation of breast cancer stem cells.

The conclusions made by the authors are generally consistent with the results. However, some points covering the current knowledge in sphingolipid biology need to be addressed in the text of the manuscript.

  1. In the current study the authors give focus to SphK1. Since the majority of cell types, including MCF-7 cell line, express both types of S1P kinases, SphK1 and SphK2, the authors should provide their arguments why only SphK1, and not SphK2, was considered for analysis. What is known about SphK2?

The same question I would raise regarding S1P receptors and S1P3 as the candidate receptor for the herein characterization.  Although in the Introduction section the authors sum up the available information about S1P3 (e.g. cross talk with the estrogen treatment, the capacity of breast cancer cells to build up mammospheres), it is not entirely clear whether other S1P receptors (may) as well play complementary roles or the information about other receptors is currently unavailable.  

  1. The authors shall provide more information and their thoughts (either in Introduction or Discussion sections) on what is known about the mechanisms of nucleo-cytoplasmic trafficking of SphK1 and S1P3 in respect of (i) nuclear localization signal and nuclear export signal, (ii) whether translocation is leptomycin B-sensitive, what in turn is indicative for a CRM1-dependent process.
  2. Diagram (B) in Figure 3 summarizes the results on S1P3 shown in Figure 4.
  3. More descriptive information should be added to the figure legend of Figure 6 to emphasize the differences for (A) and (B). In the current version, only the differential expression of S1P3 is clearly highlighted.

Minor points:

  1. Lines 16-22: The text color is grey instead of black
  2. Through the entire text: Greek symbols were not properly displayed
  3. Through the entire text: the authors use “SphK1/S1P3” when effects for the two molecules are described. This may be misleading since such type of nomenclature is often used for indication of a signaling axis. It is better to wording change to “SphK1 and S1P3”
  4. The authors should explain why for the treatment the 3 hour time point was chosen
  5. “signaling” vs. “signaling”; in the current version both writing variants are present
  6. Figure 1, A: please indicate the % of cells in each of the 4 quadrants of the dot blots
  7. Lines 246-247: the authors state: “Estrogen, a recognized growth-factor for ER-positive…”. I believe the wording needs corrections; estrogen does not belong to the classical growth factors such as EGF, VEGF, PDGF and others.

Author Response

Response to Reviewer 1

Thank you very much for your kind assessment of our work. We addressed all your questions and revised our manuscript as described below.

Major points:

 Comment 1:

In the current study the authors give focus to SphK1. Since the majority of cell types, including MCF-7 cell line, express both types of S1P kinases, SphK1 and SphK2, the authors should provide their arguments why only SphK1, and not SphK2, was considered for analysis. What is known about SphK2?

Response:

SphK1 is known to function as pro-oncogenic effector [reviewed by Sukocheva et al., 2018], whereas SphK2 has been found to mediate pro-apoptotic and growth-inhibiting effects. We have accented this fact in the Introduction section (page 2, lines 49-52, indicated in red font) and cited a relevant reference [Song et al., 2018] (reference no. 6). Considering that SphK2 nuclear trafficking has been extensively evaluated, in this study we investigated the trafficking of SphK1 and S1P3 receptor.

Comment/The same question … regarding S1P receptors and S1P3 as the candidate receptor for the herein characterization. … it is not entirely clear whether other S1P receptors (may) as well play complementary roles or the information about other receptors is currently unavailable.

Response:  

We agree with the reviewer’s opinion that we should not exclude the potential impact of other S1P receptor subtypes. We have discussed this in the Discussion section (page 9, lines 300-303; and 308-309, in red font). We indicated that out of all S1P receptor subtypes, S1P3 has the highest protein expression level in MCF-7 breast cancer cells. Previously published (numerous) findings supported the crucial role of S1P3 in regulation of cancer cell proliferation. The other S1P receptors are expressed at significantly lower levels in MCF-7 cells and were not linked to stimulation of breast cancer cell growth. The levels of S1P receptor subtypes were tested in many studies, including our own previous work [Ghosal et al., 2016; Sukocheva et al., 2009; 2013]. When more sensitive methods will be developed, future studies will address intracellular trafficking of less-represented S1P receptors in cancer cells.

Comment 2:

The authors shall provide more information and their thoughts (either in Introduction or Discussion sections) on what is known about the mechanisms of nucleo-cytoplasmic trafficking of SphK1 and S1P3 in respect of (i) nuclear localization signal and nuclear export signal, (ii) whether translocation is leptomycin B-sensitive, what in turn is indicative for a CRM1-dependent process.

Response:

According to your suggestions, the “Discussion” section has been extended, and additional 11 references have been included/cited (pages 8-9, lines 262-264, 282-293, and 315).

S1P3 nuclear trafficking remains poorly addressed. We are the first to observe regulation of S1P nuclear localization in breast cancer cells stimulated by external agents (estrogen and S1P). However, further detailed investigations are required to investigate the mechanisms of this translocation. We extended the discussion of our finding in the “Discussion” section (lines 282-285,290-293, and 315).

Comment 3:

Diagram (B) in Figure 3 summarizes the results on S1P3 shown in Figure 4.

Response:

According to your suggestion, the legend to Figure 3 has been extended to indicate the relevance of the graph to the data in Figure 4 (page 6, lines 200 and 201).

Comment 4:

More descriptive information should be added to the figure legend of Figure 6 to emphasize the differences for (A) and (B). In the current version, only the differential expression of S1P3 is clearly highlighted.

Response:

The legend to Figure 6 is now extended by providing additional explanation (page 7, lines 221-225). Sections A and B have been introduced to Figure 6 to improve the data presentation.

Minor points:

 Comment 1:

Lines 16-22: The text color is grey instead of black

Response:

We have made the correction as requested (page 1).

Comment 2:

Through the entire text: Greek symbols were not properly displayed

Response:

We have checked the manuscript and corrected/verified all Greek symbols.

Comment 3:

Through the entire text: the authors use “SphK1/S1P3” when effects for the two molecules are described. This may be misleading since such type of nomenclature is often used for indication of a signaling axis. It is better to wording change to “SphK1 and S1P3”

Response:

As suggested, all “SphK1/S1P3” have been replaced with “SphK1 and S1P3” throughout the manuscript text (indicated in red fount in the revised version of our manuscript)

Comment 4:

The authors should explain why for the treatment the 3 h time point was chosen

Response:

Preliminary time-course measurements were conducted and a 3-h exposure has been chosen as the optimal time point to reflect non-genomic trafficking responses. Longer exposure time results in genomic activation and/or irreversible transformation; for instance, during longer than 3 h of TNFa-induced activation of apoptosis, many cells start to detach, or shrink, or show signs of membrane destruction. A shorter exposure time was also investigated as a pilot study. However, the highest level of visible changes was observed at 3h. Therefore, the optimal time has been chosen to allow significant changes in non-genomic signaling (e.g., trafficking). It is now noted in the legend to Figure 3 (page 5, lines 196-197).

Comment 5:

signaling” vs. “signaling”; in the current version both writing variants are present.

Response:

As suggested, all “signalling” words have been replaced with “signaling” throughout the manuscript text (all corrections are indicated in red fount in the revised version of our manuscript).

Comment 6:

Figure 1, A: please indicate the % of cells in each of the 4 quadrants of the dot blots.

Response:

Thank you for this close observation. The Figure 1 has been revised. We inserted % numbers into two right quadrants of Figure 1. Insertion of the % on the left side will cover essential part of the lower left quadrant; therefore, it was not inserted. Insertion of smaller font numbers was not considered as useful (it will be hard to read small numbers).

Comment 7:

Lines 246-247: the authors stated: “Estrogen, a recognized growth-factor for ER-positive…”. I believe the wording needs corrections; estrogen does not belong to the classical growth factors such as EGF, VEGF, PDGF and others.

Response:

We replaced “growth-factor” with “growth-stimulating agent” (page 8, line 251, indicated in red font).

Thank you very much for this opportunity to improve our manuscript,

Kind regards,

Olga Sukocheva, PhD

Reviewer 2 Report

The manuscript adresses  an interesting and timely scientific issue related to a better understanding of SphK1/S1P3 signaling in BC cells. The manuscript expands our current knowledge to better understand intracellular traffiicking and localization of SphK1 and S1P3 in BC MCSF-7 wt and MCSF derived BCSC.enriched mammospheres trated with growth (S1P and estrogen) or apoptosis-stimulating agents (TNFa), The authors conclude and hypothesize in their study that "inhibition of nuclear/perinuclear translocation of SphK1/S1P3 represents a novel therapeutic target in breast cancer stem cells". Overall the experimental design, the results generated, the number of experimental repeats and the applied statistics are state of the art. The manuscript is well written and concise. However, the list of authors includes only one native speaker and therefore the manuscript needs some language corrections. The cited manuscripts in the Introduction and Discussion  include the most relevant literature. However, a few experimental data should be added, before the manuscript is acceptable for publication. 1) Line 122 "....and might be associated with different stages of cell growth and division." The authors have sufficient Flowcytometric expertise to adress this issue by combinig cell cycle analysis with immune-target analysis. 2) Line 162 "Live cell fluorescent monitoring....may help to spot the difference." Time kinetic analysis of 0-3h would also be helpful and should be added by the authors. 3)  Line 255 "...first limitation of the current study that phospho-SphK1 (p-SphK1) specific antibodies were not used.......". There are at least three companies that distribute commercially conjugated and un-conjugated antibodies . Therefore it is not understandable why the authors  did not include this analysis. After addition of these data under 1)-3), the manuscript is acceptable for publication.

Author Response

Response to Reviewer 2

Thank you very much for your kind assessment of our work. We addressed all your questions and revised our manuscript as described below.

Comment 1:

… Overall the experimental design, the results generated, the number of experimental repeats and the applied statistics are state of the art. The manuscript is well written and concise. However, the list of authors includes only one native speaker and therefore the manuscript needs some language corrections.

Response:

Thank you very much for an excellent assessment of our work. We addressed all your questions and revised our manuscript as described below. We have carefully checked and made a thorough editing to improve the quality of our manuscript. All corrections/ additions are indicated in red font.

Comment 2:

… a few experimental data should be added, before the manuscript is acceptable for publication.

Response:

Thank you very much for your suggestion to extend our experimental work. We are planning to continue our investigation in the field. Detailed explanations are provided to your request as indicated below for comments 3 and 5.

Comment 3:

Line 122 "....and might be associated with different stages of cell growth and division." The authors have sufficient Flowcytometric expertise to address this issue by combining cell cycle analysis with immune-target analysis.

Response:

Thank you very much for your suggestion. However, a flow cytometric analysis with commercially available anti-SphK1 antibodies require serious methodological improvements and currently is not suitable for an assessment of localization. The only antibodies that we found recommended by the manufacturer for the flow analysis are from Novus (Sphingosine Kinase 1/SPHK1 Antibody (NBP2-67164)). However, there are no publications with this reagent (https://www.novusbio.com/products/sphingosine-kinase-1-sphk1-antibody-ja31-14_nbp2-67164#reviews-publications).

The only successful detection of SphK1 by flow cytometry was performed with lymphocytes from blood samples (Li et al., 2012). Cells were then fixed with formalin and permeabilized with Cytofix/Cytoperm solution (BD Biosciences, San Jose, CA). The authors used anti-SphK1 monoclonal antibody from Cell Signaling Technology (Beverly, MA). Thus, no data validation is available to indicate their successful use in flow cytometry with breast cancer cells.

The failure may be explained by the following. Permeabilization of cells in suspension is inevitably results in some significant loss of the cytosolic enzymes and may significantly affect their intracellular localization. Accordingly, the monitoring should be performed with single live cells only as we suggested in the 2.3 Results section of our manuscript (page 4, line 164).

Comment 4:

Line 162 "Live cell fluorescent monitoring....may help to spot the difference." Time kinetic analysis of 0-3h would also be helpful and should be added by the authors.

Response:

The kinetic analysis has been performed, and 3 h exposure has been chosen as the optimal time to allow/reflect the most differences in trafficking (non-genomic effects). A shorter exposure time was also investigated as a pilot study. However, the highest level of visible changes was observed at 3h. Therefore, the optimal time has been chosen to allow significant changes in non-genomic signaling (e.g., trafficking). It is now noted in the legend to Figure 3 (page 5, lines 196-197).

Comment 5:

Line 255 "...first limitation of the current study that phospho-SphK1 (p-SphK1) specific antibodies were not used.......". There are at least three companies that distribute commercially conjugated and un-conjugated antibodies . Therefore it is not understandable the authors  did not include this analysis...

Response:

SphK1 phosphorylation has been shown to be responsible for the enzyme membrane translocation (Pitson et al., 2003; Jarman et al., 2010). We did not aim to repeat the experiments/identify membrane localization of SphK1 as it was addressed by numerous previous studies (reviewed by Siow and Wattenberg, 2011; Sukocheva et al., 2018).

As it has been previously discussed by Siow and Wattenberg (2011) that not only phosphorylated, but also palmitoylated SphK1 isoforms (other protein modifications are also possible) are associated with membranes. However, this isoform has a short half-life and low steady-state abundance on membranes (Kihara et al., 2006). Many authors indicated the low levels of membrane associated SphK1. Our experiments would not add any critical information to currently controversial findings. To address this critical issue, we have indicated the previously observed low abundance of membrane-associated SphK1 in our manuscript (page 8, lines 262-264).

Thank you very much for this opportunity to improve our manuscript,

Kind regards,

Olga Sukocheva, PhD

Reviewer 3 Report

 In this manuscript, the author showed localization of SphK1 and S1P3 in CD44+/CD24-stem cells from MCF7 cell line and parental MCF-7 cells in the presence of growth-promoting agents (S1P and estrogen) and pro-apoptotic cytokines (TNF-alpha). This manuscript covers interesting issues about the relationship between S1P and mammosphere cells; however, the manuscript has several very critical points needed to be verify.  

(1) Figure 4 and 5. The results of translocation of S1P3 from plasma membranes to nuclei and detection of S1P3 in cytosolic fraction are experiments that requires a very careful interpretation. As described by the authors in the Discussion section, S1P3 is a seven transmembrane receptor, so the potential for such proteins to translocate to the nucleus or cytoplasm is very surprising. First, the author should verify by themselves whether the S1P3 antibody used in this study really recognizes the endogenous S1P3 of the cells used, and confirm whether or not the specific bands of S1P3 in Western blotting and fluorescent signal of S1P3 is really S1P3.  To address this issue, the author should confirm specificity of S1P3 antibody by using gene knockdown or knockout experiment of S1P3. Furthermore, the author also should reconsider validity of the subcellular fractionation analysis with more organelle-specific marker proteins.   

(2)  Figure 4.  The reviewer could not see clear localization of S1P3 at plasma membranes in Control experiment. The author should perform double staining of S1P3 and plasma membrane by using such as FM4-64 and should confirm the colocalization.   

(3)  Figure. 3B.  Figure 3A shows localization of SphK1, but not S1P3. However, the graph of Figure 3B contains quantification of results of both SphK1 and S1P3.  The quantification results of S1P3 should move to Figure 4. 

(4)  Figure 6. Line 164 “although the level of S1P3 expression was higher compared to parental MCF-7 wild-type cells”. Since the immunofluorescence data of BCSC and parental MCF7 cells exist on the separated figure, it is not possible to make a comparison like the previous sentence. 

(5)  In this manuscript, there is no experimental evidence that links Sph1K1 or S1P3 translocation to cell proliferation or apoptosis. The author examines effects of growth-promoting agents and pro-apoptotic cytokines on the translocation (Figure 3-5), but this does not provide experimental evidence that the translocation affects growth or apoptosis.

Author Response

Response to Reviewer 3

Thank you very much for your kind assessment of our work. We addressed all your questions and revised our manuscript as described below.

Comment 1:

Figure 4 and 5. … the potential for such proteins to translocate to the nucleus or cytoplasm is very surprising. First, the author should verify by themselves whether the S1P3 antibody used in this study really recognizes the endogenous S1P3 of the cells used, and confirm whether … signal of S1P3 is really S1P3.  To address this issue, the author should confirm specificity of S1P3 antibody by using gene knockdown or knockout experiment of S1P3. Furthermore, the author also should reconsider validity of the subcellular fractionation analysis with more organelle-specific marker proteins.

Response:

Indeed, we have detected translocation of S1P3 receptors to the nuclei of breast cancer cells stimulated with external agents (estrogen and S1P), although we are not the first to report similar observation of nuclear trafficking of transmembrane G-protein coupled receptors (GPCR). Nuclear translocation of GPCR has recently been reviewed by Jong et al., 2018.

Several mechanisms of nuclear GPCR translocation have been reported. These range from lateral diffusion through peripheral channels between the nuclear pore complex and the pore membrane to movement through the nuclear pore complex using linkers, carrier proteins and even components of the soluble transport machinery (Jong et al., 2018). Some GPCR contains nuclear localization signals (NLSs), the short stretches of basic amino acids that are subsequently recognized by specific members of the karyopherin superfamily for nuclear import (reviewed by Jong et al., 2018).

Notably, it was reported that SphK1 contains two functional nuclear export signal (NES) sequences (Inagaki et al., 2003). We cited this information in our revised manuscript (page 8, lines 283 and 284). We also mentioned in the Discussion section that several other GPCRs were recently found in the nuclear space, suggesting novel roles for this type of receptors beyond those traditionally established (page 9, line 315).

Regarding your concerns about the antibody validation, we note that antibody validation has been performed by the manufacturer. Furthermore, we validated the specificity of the used anti-SphK1 and anti- S1P3 antibodies in our previous work using recombinant proteins (Sukocheva et al., 2009; Sukocheva et al., 2013) (page 3, line 127).

Comment 2:

Figure 4.  The reviewer could not see clear localization of S1P3 at plasma membranes in Control experiment. The author should perform double staining of S1P3 and plasma membrane by using such as FM4-64 and should confirm the colocalization.

Response:

Plasma membrane -localization of S1P3 is a very dynamic process as there is a lot of S1P in plasma/extracellular fluids which as been mentioned in our manucript (page 2, lines 64-68). Therefore membrane-localized receptors are in constant movement from the membrane to cytosol due to S1P3 internalization after S1P binding. Our previous publication addressed this process in full details (Sukocheva et al, 2013) (page 2 and 3, lines 69 and 127). The plasma membrane localization of S1P3 is an established fact that does not require further verification.

Comment 3:

Figure. 3B.  Figure 3A shows localization of SphK1, but not S1P3. However, the graph of Figure 3B contains quantification of results of both SphK1 and S1P3.  The quantification results of S1P3 should move to Figure 4.

Response: According to your suggestion, the legend to Figure 3 has been expanded to indicate the relevance of the graph to the data in Figure 4.

Comment 4:

Figure 6. Line 164 “although the level of S1P3 expression was higher compared to parental MCF-7 wild-type cells”. Since the immunofluorescence data of BCSC and parental MCF7 cells exist on the separated figure, it is not possible to make a comparison like the previous sentence. 

Response:

This conclusion/assumption (the level of S1P3 expression is higher in mammospheres compared to parental MCF-7 wild-type cells) is supported by the Rt-PCR data (Figure 2, panel C). This information is now incorporated in our revised manuscript (page 4, lines 167 and 168).

Comment 5:

… there is no experimental evidence that links Sph1K1 or S1P3 translocation to cell proliferation or apoptosis. The author examines effects of growth-promoting agents and pro-apoptotic cytokines on the translocation (Figure 3-5), but this does not provide experimental evidence that the translocation affects growth or apoptosis.

Response:

Thank you very much for this thought-provoking comment. It is an appropriate direction for future investigations. To address the mechanism of translocation, it is necessary to identify the trafficking effectors. Currently, it is unclear what kind of inhibitors to use to link the translocation to growth/apoptosis. In the future studies, the next step would be to identify the translocation-involved protein, inhibit it, and see if the proliferation will be also blocked. Previous studies indicated a striking localization of endogenous SphK1 with centrosomes in HEK293 cells (Gillies et al., 2009). We presented this in the “Discussion” section (page 8, lines 264-286).

We also like to bring to your kind notice that we do not claim a direct link between growth/apoptosis and the detected trafficking. As indicated in our “Discussion” section, our current study detected differences in the SphK1 and S1P3 localization in parental and MCF-7-derived BCSC-enriched mammosphere cells treated by growth- and apoptosis-inducing agents (page 10, lines 361-369).

Thank you very much for this opportunity to improve our manuscript,

Kind regards,

Olga Sukocheva, PhD

Round 2

Reviewer 2 Report

The authors satisfactorily responded to all reviewer comments and significantly improved the manuscript. the manuscript is now acceptable for publication.

Reviewer 3 Report

This manuscript is acceptable.